# Nitrated Graphene Oxide Derived from Graphite Oxide: A Promising Energetic Two-Dimensional Material

**DOI:** 10.3390/nano11010058

**Published:** 2020-12-29

**Authors:** Fayang Guan, Hui Ren, Lan Yu, Qingzhong Cui, Wanjun Zhao, Jie Liu

**Affiliations:** 1State Key Laboratory of Explosion Science and Technology, Beijing Institute of Technology, Beijing 100081, China; casguan@163.com (F.G.); cqz1969@bit.edu.cn (Q.C.); wanjunzhaowj@gmail.com (W.Z.); liujie0417@bit.edu.cn (J.L.); 2Hong Kong New ARK Technologise Ltd., Hong Kong 999077, China; gutti14@126.com

**Keywords:** nitrated graphene oxide, nitrification process, properties analysis, synchronous simultaneous analysis, combustion performance

## Abstract

In order to synthesize a novel two-dimensional energetic material, nitrated graphene oxide (NGO) was prepared by the nitrification of graphite oxide to make a functional modification. Based on the morphological characterization, the NGO has a greater degree of curl and more wrinkles on the surface. The structure characterization and density functional theory calculation prove that epoxy and hydroxyl groups on the edge of graphite oxide have reacted with nitronium cation (NO_2_^+^) to produce nitro and nitrate groups. Hydrophobicity of NGO implied higher stability in storage than graphene oxide. Synchronous simultaneous analysis was used to explore the decomposition mechanism of NGO preliminarily. The decomposition enthalpy of NGO is 662.0 J·g^−1^ and the activation energy is 166.5 kJ·mol^−1^. The thermal stability is similar to that of general nitrate energetic materials. The hygroscopicity, thermal stability and flammability of NGO prove that it is a novel two-dimensional material with potential applications as energetic additives in the catalyst, electrode materials and energetic devices.

## 1. Introduction

Graphene oxide (GO) is an amphiphilic two-dimensional material generally exfoliating from graphite oxide [1,2,3,4]. Many models on GO such as the Lerf–Klinowski structural model [5,6,7], the dynamic structural model [8] and the two-component structural model [9] reveal that the majority of oxygen functions in GO are epoxy and hydroxyl with few carboxyl and carbonyl existing on the edge [10,11]. These groups are active and could be modified to obtain different materials: the carboxyl group reacts with thionyl chloride to produce high-efficiency photovoltaic materials [12]; N-vinylcarbazole could grow on the surface by the hydroxyl group [13]; amine nucleophiles undergo ring-opening reaction with epoxy groups [14]. Meanwhile, GO has been regarded as a promising two-dimensional energetic material in catalysis and energy storage fields. However, the thermodynamic property of GO is unstable to undergo exothermic disproportionation reactions readily [15], and these hydrophilic oxygen-groups lower the storage behavior of GO. To solve these problems and realize the energetic functionalization of graphene sheets, it is necessary to modify the oxygen-containing groups of GO.

Nitrification is widely used in organic reactions. In 1834, benzene was nitrated into nitrobenzene [16,17]. Nitro is the energetic functional group of most explosives. In the energetic materials, the nitro group connects with a carbon atom like in 2,4,6-trinitrotoluene (TNT), attaches to oxygen atom like in nitrocellulose and links with nitrogen atom like in cyclotrimethylenetrinitramine. The direct nitrification of graphene is hard to realize. The nitrification reagent is an overly strong oxidation agent that would produce oxygenated groups on GO instead of being attached as NO_2_ group. The reagent after oxidation reaction is unable to nitrate GO afterward. Therefore, modification of GO to prepare nitrated graphene oxide (NGO) is a facile approach to make GO more energetic. NGO is a promising energetic additive in explosive systems. It can activate the reaction of the energetic system, lower the critical threshold, and release heat more quickly. Zhang [18] and Yuan [19] added NGO into energetic materials as catalysis with the decrease in the thermal decomposition temperature. Nevertheless, the mass content of nitrogen in NGO is only 1.45%, which prohibits energetic behavior. Thus, the research of the nitrification process is of great significance for increasing the nitro content of NGO. The analysis of hydrophilic properties, thermal stability and combustion performance could explore the application prospect of NGO as an energetic addictive.

In this paper, the mixture of nitrate, glacial acetic acid and concentrated sulfuric acid was used to nitrate GO. The mixed acid could protonate the nitrate ion and subsequently make water elimination, which could produce nitronium cation (NO_2_^+^) with high concentration effectively and safely. Then the nitrification reaction process was determined by the structure analysis and density functional theory (DFT) calculation. The dispersion ability of the NGO in solvents was explored and molecular dynamics (MD) was employed to simulate the changes in hydrophilicity, reflecting the hygroscopicity of NGO indirectly. Simultaneous analysis measurements were used to analyze the thermal stability and the combustion performance of NGO has been observed preliminarily.

## 2. Experimental Section

### 2.1. Calculation Methods

In order to investigate the reaction process of NO_2_^+^ and GO, A simplified model of GO that only includes epoxy and hydroxyl was built, which is shown in Figure 1. Electrostatic potential and Fukui function [20] of GO were calculated. Different distances between GO and NO_2_^+^ were constrained to analyze the changes in the structure (both bond length and bond angles) when the reactants get closer. The DFT results were obtained by using the DMol3 density-functional module in the set of general gradient approximation (GGA) functionals and Perdew–Burke–Ernzerhof (PBE) correlation [21]. We set 1 × 10^−5^ Ha as the convergence tolerance, DNP 3.5 was the basis set and 0.002 Ha of smearing was used to speed up the convergence.

MD was used to simulate the hydrophilic of GO and NGO. 500 water molecules were put into a cell (33.44 × 33.44 × 33.44 Å^3^) and the GO and NGO models were established in the results of the element analysis as shown in Figure 2. The simulation was carried out by the Forcite module in the COMPASS forcefield. A fixed time step of 1 fs was used in all cases and the whole runs of 1 ns duration were performed in the NPT system. Radial distribution function (RDF) and interaction energies (*E*_inter_) were calculated by the result of dynamic simulation.

Dmol3 module and Forcite module were implemented in the Materials Studio 5.5 from Accelrys Inc.

### 2.2. Sample Preparation

Graphite oxide was prepared by the Hummers method [22] and the brief process was showed as following. 3.5 g graphite was mixed with 120 mL concentrated sulfuric acid (H_2_SO_4_, AR) in a flask. Then, 3.5 g sodium nitrate (NaNO_3_, AR) was added slowly. The temperature of the steps should be ensured under 10 °C and the solution needs to be stirred continuously. We added 15 g potassium permanganate (KMnO_4_, AR) into the mixture to induce a reaction for 4.5 h in 35 °C. Deionized water was used to dilute the solution and 40 mL hydrogen peroxide (H_2_O_2_, concentration 30%) was employed to react with the remaining KMnO_4_. Hydrochloric acid (HCl 1 mol/L) and acetone (CH_3_COCH_3_ AR) was used to remove the inorganic impurities [23,24]. Then graphite oxide was obtained after drying in the vacuum oven. 1 g graphite oxide and 10 g NaNO_3_ were ultrasonically dispersed in 150 mL of glacial acetic acid (CH_3_COOH, AR). 70 mL concentrated H_2_SO_4_ was dripped into the mixture under the ice bath. Then the temperature was raised to 45 °C, and the mixture was stirred for 6 h. The nitrated graphite was prepared after diluting, cleaning and drying.

Deionized water, dimethylformamide (DMF, AR), N-methylpyrrolidone (NMP, AR) and o-dichlorobenzene (o-DCB, AR) were employed as the solvents. Taking the deionized water as an example: 100 mg nitrated graphite was dispersed into 300 mL deionized water, and then was exfoliated by ultrasonic crusher under the power of 250 W for 9 h. The suspension was centrifuged at 1500 rpm for 2 h and the NGO with few layers remained in the supernatant. GO dispersion in these solvents were prepared in the same way. NGO and GO aqueous dispersions were dried in the vacuum freeze oven to obtain solid samples. The entire preparation process is shown in Figure 3.

Graphite (D_90_ = 40 µm, 99.999%) was purchased from Nanjing XFNANO Materials Tech Co., Ltd., Nanjing, China. Deionized water was made in the laboratory and other reagents were all from Aladdin Industrial Co., Shanghai, China.

### 2.3. Materials Characterization

Scanning electron microscopy (SEM, S-4700, HITACHI, Chiyoda, Japan), transmission electron microscopy (TEM, JEM-2010, HITACHI, Chiyoda, Japan) and atomic force microscopy (AFM, Dimension Icon, BRUKER, Karlsruhe, Germany) were used to characterize the morphology of the samples. Fourier transform infrared spectrometer (FTIR, VERTEX70, BRUKER, Karlsruhe, Germany), nuclear magnetic resonance spectroscopy (13C-NMR, CMX 400, VARIAN, Palo alto, CA, USA), Raman spectrometer (RM1000, Renishaw, Gloucester, England) and organic element analyzer (EA3000, Euro Vector, Pavia, Italy) were carried out to analyze the structure.

### 2.4. Thermal Analysis

Differential scanning calorimetry (DSC) and thermogravimetry (TG) were used to investigate the thermal decomposition of energetic materials. Combined with the measurements of infrared spectroscopy (IR) and gas chromatography (GC) or mass spectrum (MS) for analyzing the gaseous products, the process of decomposition can be initially explored. Multi-apparatus combination as DSC-TG-MS-FTIR attracts increasing interest due to its synchronous analysis of gaseous products for improving accuracy compared to an individual combination [25,26,27]. In this paper, DSC-TG-MS-FTIR synchronous measurement is an assembly of NETZSCH (Bayern, Germany) STA449C, NETZSCH QMS403C and BRUKER (Karlsruhe, Germany) VERTEX 70, and two pipelines are connected with DSC/TG measurements to MS apparatus and FTIR apparatus separately. The details about the connection could be taken in Ref. [26]. NGO was tested from room temperature to 360 °C under an argon atmosphere at a heating rate of 20 °C·min^−1^ in simultaneous thermal analysis.

### 2.5. Combustion Properties

500 mg of the NGO powder was dispersed in 100 mL of water to prepare colloids, then colloids were filtered and dried to prepare the NGO sheet. Several strips with complete shape and uniform size (60 mm × 5 mm) were cut from the NGO slice for the combustion experiment.

In order to simulate the function of gas work, 10 mg of the NGO powder prepared under vacuum freeze-drying conditions was weighed to press out a 10 mm diameter NGO tablet and put in the bottom end of the syringe. The small hole at the front of the syringe was sealed by PTFE tape and the NGO was heated by an alcohol burner. The value was determined after the tube cooling to room temperature.

A high-speed video camera (FASTCAM-APX RS, Photron, Tokyo, Japan) was used to capture the photography images of combustion performance and gas work process at the imaging frequency of 125 fps.

## 3. Results and Discussion

### 3.1. Morphological Analysis

As shown in Figure 4a,b, more wrinkles and bends exist on the surface of NGO than GO. The integrity of the graphene lattice may be damaged by the nitrification process. These uneven surfaces are able to expand contact area with other materials, beneficial to heat transformation. The uneven surface is also shown in TEM images (Figure 4c). The high-resolution image of the edges of NGO sheets indicates that the NGO sheet has only two layers. In AFM images (Figure 4d), we measure the thickness of the NGO sheet, which is around 1.1 nm. Thirty samples are counted for thickness and the values are all the multiple of 1.0–1.1 nm. Only three samples were given over five layers, which could prove low power stripping for a long time could exfoliate graphene materials well.

### 3.2. Structure Analysis

In the element analysis, the mass fraction of GO and NGO are 68.8/30.6/0.6 (C/O/H) and 57.6/8.0/33.9/0.5 (C/N/O/H) respectively. After simplifying to the chemical formula, the NGO is C_48_H_5_O_21.2_N_5.7_ and GO is C_47.8_H_5_O_15.9_. In the GO sheet, the number ratio of O/C is about 0.33 and the value would increase to 0.44 after nitrification.

The FTIR spectra of NGOs could be observed in Figure 5a. The peaks at 1581 cm^−1^ and 1365 cm^−1^ correspond to the symmetry and antisymmetric stretching vibration of nitro groups (–NO_2_). These two peaks were wider with higher wavenumbers than other nitro compounds. It means the nitro groups may connect to other atoms. The intensity of stretching vibration of –OH (3421 cm^−1^), Ar–O (1280 cm^−1^), C–OH (1089 cm^−1^), O–C–O (1047 cm^−1^), and bending vibration of –OH (1390 cm^−1^) become weaker after nitrification. The 13C NMR spectra (Figure 5b) of NG has been used to determine the nitrogen-contained functional group. The graphene lattice is all sp^2^ hybrids, and the chemical shift in the range from 120 ppm to 130 ppm would be regarded as graphene lattice [28,29]. Two peaks located at 148 ppm and 157 ppm have been detected that are corresponding to the nitro group and nitrate group respectively. The intensity of donor groups (C–OH at 68 ppm and C–O–C at 57 ppm) drop significantly in the NGO diagram. Characteristic peaks of epoxy and hydroxyl both in FTIR spectra and 13C NMR spectra are greatly reduced. The content of carbonyl has a weak change. From the analysis of FTIR spectra and 13C NMR spectra, GO was nitrated to NGO successfully.

At the wavelength of 514 nm, the graphene of Raman spectra has two characteristic peaks: G peak of graphene at 1580 cm^−1^ that is unique to the carbon sp^2^ structure, which can reflect the crystal integrity. D peak near 1350 cm^−1^ is a defect peak, reflecting the disorder of the graphene sheet. The intensity ratio (*I*_D_/*I*_G_) is usually used to characterize the degree of covalent modification of graphene, and the ratio rises from 0.93 to 1.07 after nitrating as demonstrated in Figure 5c. The reaction further disrupts the order of the carbon lattice, which might cause the deformation of the layer structure as the SEM images shown in Figure 4b.

### 3.3. DFT Calculation Analysis

From the structure analysis above, we could know the epoxy and hydroxyl groups reacted with NO^2+^ to produce C–NO_2_ and O–NO_2_ groups. However, the reaction process cannot be reflected. In order to research the nitrification, a simplified GO model only consists of epoxy and hydroxyl groups were built, which is shown in Figure 1b. The calculation result of the electrostatic potential diagram shows that the density of the electron cloud at the oxygen atoms is greater, and these sites attract cations more easily. Fukui function is the index to assess a priori reactivity of chemical species from their intrinsic electronic properties. The electrophilic Fukui function *f* − (*r*) is used to measure the sensitivity of the reaction with the electrophilic attack as shown in Equation (1).
*f* − (*r*) = [ρ*_N_*(*r*) − ρ*_N+∆_*(*r*)]/_∆_*N,*(1)
ρ(*r*) is the charge density and ∆*N* represents the changes of electrons.

For the convenience of describing each atom, these function groups have been labeled, shown in Figure 1b. O_3_ is the oxygen atom of epoxy in the center of NGO and O_4_ is the oxygen atom of epoxy in the edge. O_5_ and O_6_ are oxygen atoms of hydroxyl in the edge and center of NGO respectively. The results in Table 1 show that the epoxy and hydroxyl groups on the edge are more vulnerable to electrophilic attacks than the groups on the surface.

Changes in atomic structure can react to the specific nitrification process indirectly. The hybrid type of nitrogen atom in NO_2_^+^ turns from sp to sp^2^ when it bonds with other atoms. The bonding atoms centered on N tend to be on the same plane. Hence, a facile method could be used to further determine the reaction type and sites by analyzing the changes of bond length, angles and coplanarity in the stable configuration when groups and NO_2_^+^ are at different distances. As shown in Appendix A, as the distance between epoxy groups and NO_2_^+^ gets closer, NO_2_^+^ and the oxygen atoms of epoxy tend to be coplanar. However, both carbon-oxygen bonds of the central epoxy group become longer, and then break when N atom and O_3_ atom are in the bonding distance. The marginal epoxy only fractures one carbon-oxygen bond in 1.63 Å of *L*(N··O_4_) (the length between N atom and O_4_ atom, ‘··’ means unbonded function and ‘–’ is the bond function). It means that the epoxy groups on the edge can undergo a ring-opening reaction to form a nitrate group. In Appendix A, when NO_2_^+^ gets closer to the marginal hydroxyl group, the coplanarity of NO_1_O_2_C_5_ gets better and *L*(O_5_–C_5_) becomes longer, tending to form the nitro group. Meanwhile, the changes in NO_1_O_2_O_5_ and *L*(O_5_–H) have the same tendency. This analysis illustrates that these two groups could take esterification and substitution reactions. Therefore, when the *L*(N··O_5_) and *L*(N··C_5_) get shorter to bond, the configuration is hard to stabilize. For the result of central hydroxyl and NO_2_^+^ in Appendix A, NO_1_O_2_O_6_ tends to be in a plane, but the O_6_–H bond is stable firmly. The coplanarity of NO_1_O_2_C_6_ is fine outside the C··O_6_ bonding distance because of the conjugate between NO_2_^+^ and graphene base. However, NO_2_^+^ prefers to connect with hydroxyl rather than forming a nitro group when O_6_–C_6_ bond breaking. From the analysis above, NO_2_^+^ could react with the epoxy and hydroxyl groups on the edge in accordance with the result of the Fukui function index.

### 3.4. Dispersion and Hydrophobicity

The changes in structure affect dispersion performance. Since the preparation methods of GO and NGO dispersion are the same, the affinity of them with different solvents can be compared by the concentration of the dispersion. The result (Figure 6a) shows that NGO is not as dispersible as GO in an aqueous solution. The dispersibility of NGO in o-DCB is better than that of GO, and NMP could be regarded as an excellent solvent, which is consistent with the hydrophobicity of nitro.

The hygroscopicity is an important index for an energetic addictive and MD simulation was used to analyze the hydrophobicity performance of GO and NGO. RDF (*g*(*r*), also sometimes referred to as pair correction function [30]) is the probability of atoms that, given the presence of an atom at the reference frame, there will be an atom with its center located in a spherical shell of infinitesimal thickness at a distance, *r* (Å).

In the results of the RDF diagram (Figure 6b), the starting values are the same (0.85 Å), but the peak of the NGO/water system is lower than the GO/water system. It means the addition of these two-dimensional materials cannot strengthen the interatomic force, but the atomic aggregation in the NGO system is weaker than the GO system. The addition of GO can absorb more water molecules than NGO. Hence, NGO behaves worse hydrophilicity than GO, which is consistent with the concentration results. It means that NGO is expected to have better storage performance and a longer life span than GO.

### 3.5. Thermal Analysis

The thermal performance is an indispensable premise for basic theoretical research such as catalysis, combustion and detonation [31,32,33,34,35]. From the TG/DSC spectra in Figure 7a, the decomposition of NGO has a major exothermic peak in 218 °C with a 48% mass loss. Enthalpy of decomposition (Δ*H*_d_) could be obtained by integrating the peak area as 662 J·g^−1^ (negative values indicate exotherm). This level is between the nitroglycerin (353.8 J·g^−1^ [36]) and pentaerythritol tetranitrate (881 J·g^−1^ [37]). Element analysis of NGO and solid remains illustrates that about 4.6% of carbon atoms as well as the whole N, O, and H atoms participate in the decomposition reaction to generate gas products. These gas fragments were measured by synchronous MS analysis. There are two possibilities for the *m/z* of 44, namely N_2_O^+^ and CO_2_^+^. The fragment could be determined by the synchronous FTIR analysis that the spectra of gas have characteristic peaks of CO_2_^+^ (two peaks at 2312 cm^−1^ and 2368 cm^−1^ shown in Figure 7b). Then the reaction mechanism of NGO could be speculated preliminarily: the decomposition of oxygen-containing groups produces CO_2_^+^ and H_2_O^+^; and NO_2_^+^ is from the break of nitro groups; HNO_2_^+^ may come from the hydrogen atom rearrangement of nitrate groups and then take disproportionation to produce NO^+^.

To obtain the corresponding kinetic parameters (apparent activation energy *E*_a_, pre-exponential factor *A*), DSC and TG curves at the heating rates of 5, 10, 15 and 20 °C·min^−1^ were dealt with by mathematic means, and the Kissinger [38] method were employed as shown in Equation (2).
ln(*β*/*T*_p_^2^) = ln(*A*R/*E*_a_) − (*E*_a_/R*T*_p_),(2)
where *T*_p_ is the peak temperature (°C), *A* is the pre-exponential factor (s^−1^), *E*_a_ is the apparent activation energy (kJ·mol^−1^), β is the heating rate (°C·s^−1^), R is the gas constant (8.314 J·mol^−1^·°C^−1^). The results are shown in Table 2.

It can be seen from the experimental data (Table 2) that the *T*_p_ of NGO ranges from 201.4 to 218.0 °C, and the Δ*H*_d_ increases as the *β* get larger. The *E*a of NGO is 166.6 kJ/mol in the range of 163 kJ/mol and 188 kJ/mol [39,40,41], which is the first-order (O–NO_2_ bond fracture) reaction energy of several nitrate energetic materials. This further confirms that the initial thermal decomposition of NGO is the breaking of O–NO_2_ bond, and NGO is equivalent to common nitrates in thermal stability and heat release. Compared to the decomposition performance of GO at the heating rate of 10 °C [42], the exothermic peak of NGO is 209.7 °C higher than that of GO around in 208.7 °C slightly.

### 3.6. Energy Response

The thermal stability of GO so low that even can explode upon being heated due to the remaining potassium salt in the preparation. The GO containing potassium salt will rapidly disproportionate and generate a large amount of water vapor and CO_2_ gas with heat release with the catalysis of potassium salt. However, the purified GO cannot be ignited, which might be used as a flame-retardant material [15]. Thus, although the non-energetic GO could be used as an additive to reduce the sensitivity of energetic materials [43,44,45,46,47], GO would reduce the energy-out.

Therefore, an energetic NGO would be a more promising additive when compared to GO. After being nitrated and using the same cleaning method to remove metal ion impurities, NGO tends to combust violently in Figure 8a. The NGO trip with 60 mm length could be ignited within 20 ms and burned off in 400 ms with intense fire. The average burning rate was around 0.15 m/s. In the gas workability experiment, 10 mg of the NGO powder could generate about 5.8 mL gas finally. NGO performs better combustion behavior and rapid ignition respond with a certain amount of gas products to do work. The combustion tests show that when NGO is used as an additive for energy, it could enhance the combustion performance and energy response of the system.

## 4. Conclusions

The NGO was prepared by the nitrification of graphite oxide and the NO_2_^+^ prefer to react with the epoxy and hydroxyl groups on the edge to produce C-NO_2_ and O-NO_2_ groups. Structural changes cause changes in performance. The NGO behaves hydrophobicity, similar thermal property with other nitrate materials and fine combustion performance. These properties might conduce to storage and promote reaction in the energetic formula. Therefore, NGO is a promising two-dimensional material as an energetic additive in catalysts, electrode materials and other fields.

## Figures and Tables

**Figure 1 nanomaterials-11-00058-f001:**
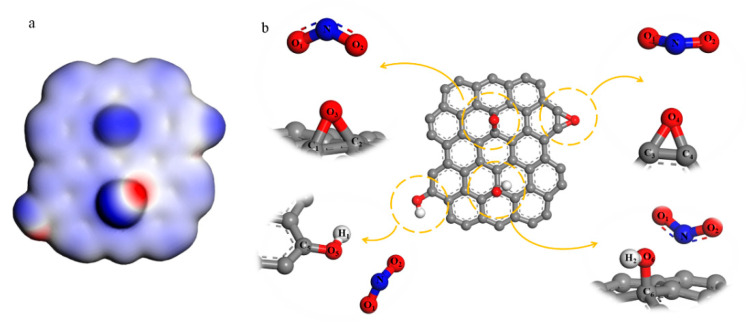
(**a**) Electrostatic potential diagram of simplified graphene oxide (GO) model; (**b**) the reaction between NO^2+^ and hydroxyl and epoxy groups both in the surface and the edge.

**Figure 2 nanomaterials-11-00058-f002:**
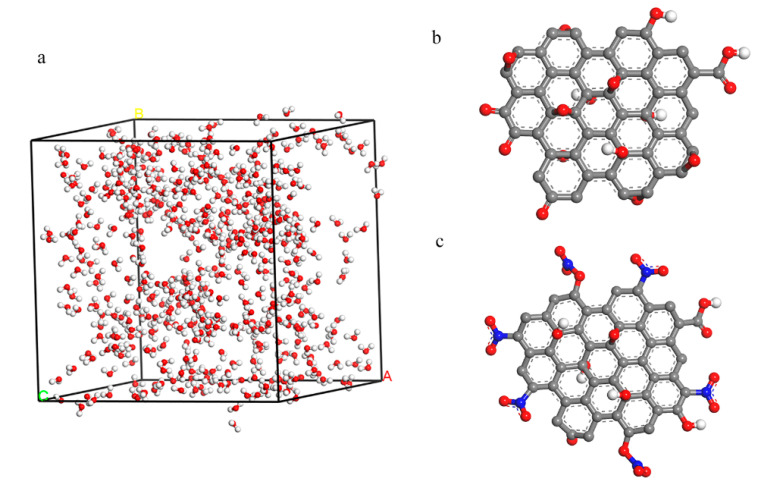
(**a**) The 500 water models inside the cell with a size of 33.44 × 33.44 × 33.44 Å^3^; (**b**) the model of GO; (**c**) the model of nitrated graphene oxide (NGO).

**Figure 3 nanomaterials-11-00058-f003:**
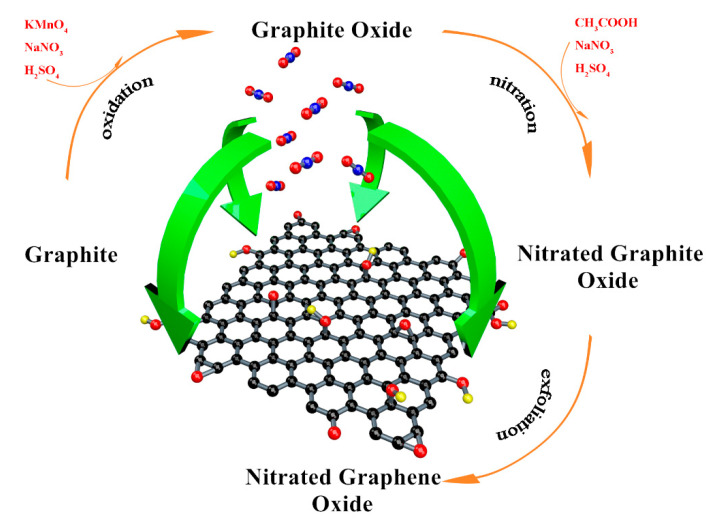
The entire preparation process of NGO.

**Figure 4 nanomaterials-11-00058-f004:**
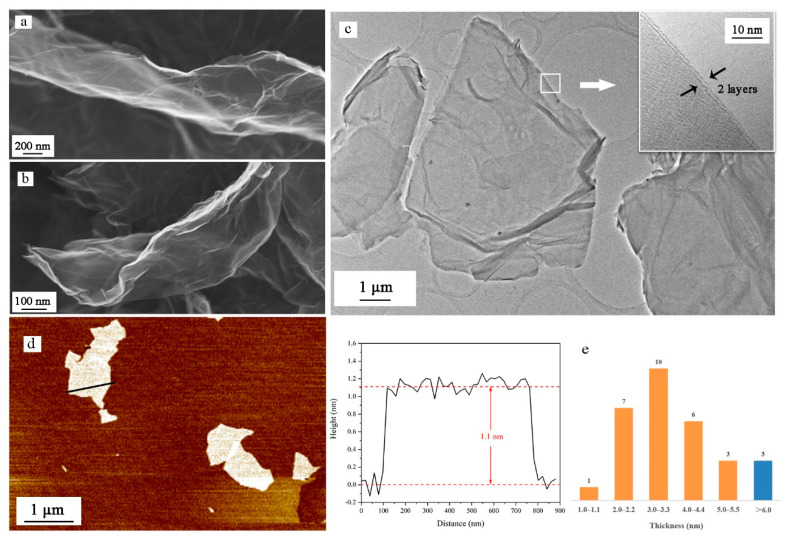
(**a**) Scanning electron microscopy (SEM) images of GO sheet; (**b**) SEM images of NGO sheet; (**c**) transmission electron microscopy (TEM) images of NGO sheet and high-resolution imagine of the edges of NGO; (**d**) atomic force microscopy (AFM) images of NGO sheet and the height distribution along the path; (**e**) thickness distribution of 30 NGO samples.

**Figure 5 nanomaterials-11-00058-f005:**
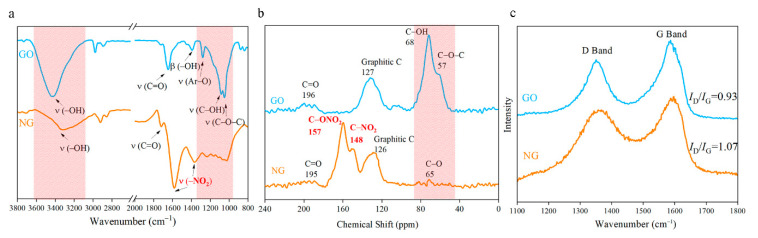
(**a**) Fourier transform infrared spectrometer (FTIR) spectra of NGO and GO; (**b**) 13C nuclear magnetic resonance spectroscopy (NMR) spectrum of NGO and GO; (**c**) Raman spectra of NGO and GO.

**Figure 6 nanomaterials-11-00058-f006:**
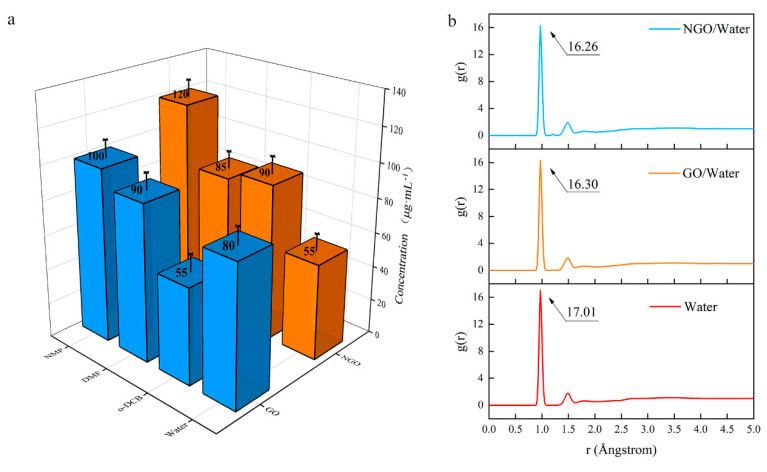
(**a**) The concentration of GO and NGO in different solvents; (**b**) The radial distribution function (RDF) diagram of water, GO/water system and NGO/water system.

**Figure 7 nanomaterials-11-00058-f007:**
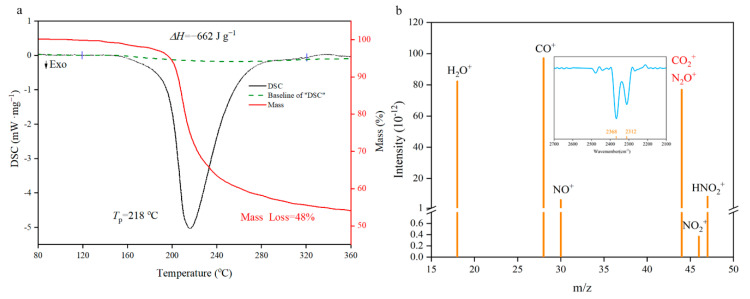
Differential scanning calorimetry (DSC)-thermogravimetry (TG)-mass spectrum (MS)-FTIR characterization diagrams. (**a**) The TG/DSC spectra of NGO; (**b**) the MS intensity spectra of gas fragments with *m*/*z* as 18(H_2_O^+^), 28(CO^+^), 30(NO^+^), 44(N_2_O^+^ or CO_2_^+^), 46(NO_2_^+^) and 47(HNO_2_^+^). IR spectra in the range of characteristic wavelength shown on the top.

**Figure 8 nanomaterials-11-00058-f008:**
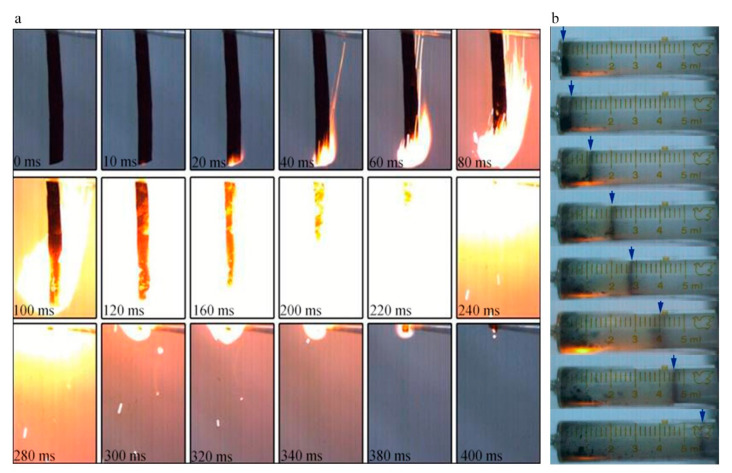
(**a**) The combustion images of NGO; (**b**) the photographs of work experiment.

**Table 1 nanomaterials-11-00058-t001:** Mulliken atomic charges and Fukui function of oxygen atoms.

	O_3_	O_4_	O_5_	O_6_
Mulliken atomic charges	−0.351	−0.389	−0.402	−0.401
*f* − (*r*)	0.017	0.031	0.024	0.018

**Table 2 nanomaterials-11-00058-t002:** Thermal analysis data.

*β* (°C·s^−1^)	*T*_p_ (°C)	Δ*H*_d_ (J·g^−1^)
5	201.4	425
10	209.7	473
15	213.7	577
20	218.0	662
*E*_a_ = 166.6 kJ/mol	lg*A* = 6.10	R_k_ = 0.9921

## Data Availability

Data is contained within the article or supplementary material. The data presented in this study are available in supplementary material.

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
