# Peer review of "Nitrated Graphene Oxide Derived from Graphite Oxide: A Promising Energetic Two-Dimensional Material"

_nanomaterials, 2020, doi:10.3390/nano11010058_

Round 1

Reviewer 1 Report

The authors of this study replied to all my comments in a satisfactory way.

From my point, the manuscript can now be accepted and published.

Author Response

Thanks for the reviewer's comments and suggestions. They have great help to improve the quality of the article.

Reviewer 2 Report

The authors have clearly responded all the issues raised by the reviewers and made appropriate changes in the revised manuscript. Therefore, the manuscript can be accepted for publication. 

Author Response

Thanks for reviewer's comments and suggestions. They have great help to improve the quality of the article.

Reviewer 3 Report

The authors responded well to the reviewer's questions, refuting my objections where it was justifiable, and changing the points where they needed to make corrections. 

I think the paper is publishable in Nanomaterials, if the following minor points are considered by the authors (no further review looks necessary from my side):

- "Nitrification reagent would behave strong oxidizing property" is formulated wrongly. I get the point what it might mean, but you need to reformulate e.g. to "the nitrification reagent is an overly strong oxidation agent that would produce oxygenated groups on GO instead of being attached as NO2 group".

- "And then the concentration of reagent is unable to nitrate GO
afterwards", is also weakly formulated, because the "concentration" is not unable to anyting, but it is the reagent itself, so I suggest writing "the reagent in the applied contration is unable to nitrate GO afterwards"

- The authors write "Due to the stable
structure of graphene sheet, the direct nitrification of graphene is hard to realize.". I really do not know if this is a well-based argument. Probably it is, because the stability of graphene sheet is indeed stable. That is true. However, the authors also write that benzene is also nitrated, and this is also right. Benzene is also a stable material, but under harsh conditions is undergoes nitration, while graphene does not. This is not a critical point, though, I think that while I have doubts, I can still suggest to leave the question open and the paper being published.

- another minor suggestion: "nitro connects" needs correction, because it is a "nitro group".

Author Response

We appreciate very much for your positive and constructive comments and suggestions on our manuscript. Point by point responses to the comments have been listed below.

Reviewer 3:

- "Nitrification reagent would behave strong oxidizing property" is formulated wrongly. I get the point what it might mean, but you need to reformulate e.g. to "the nitrification reagent is an overly strong oxidation agent that would produce oxygenated groups on GO instead of being attached as NO2 group".

Response: It’s our negligence for inaccurate writing, and we have changed the expression in Page 2 Line 7. Delete ‘Nitrification reagent would behave strong oxidizing property to destroy the graphene sheet and produce oxygen containing groups preliminarily’ and add ‘The nitrification reagent is an overly strong oxidation agent that would produce oxygenated groups on GO instead of being attached as NO2 group’.

- "And then the concentration of reagent is unable to nitrate GO afterwards", is also weakly formulated, because the "concentration" is not unable to anyting, but it is the reagent itself, so I suggest writing "the reagent in the applied contration is unable to nitrate GO afterwards"

Response: It’s our negligence for inaccurate writing, and we have changed the expression in Page 2 Line 9. Delete ‘And then the concentration of reagent is unable to nitrate GO afterwards’ and add ‘And the reagent after oxidation reaction is unable to nitrate GO afterwards’. The suggested sentence from reviewer has a little mistake (word ‘contration’ is wrong), and we change it into another version without changing the meaning.

- The authors write "Due to the stable structure of graphene sheet, the direct nitrification of graphene is hard to realize.". I really do not know if this is a well-based argument. Probably it is, because the stability of graphene sheet is indeed stable. That is true. However, the authors also write that benzene is also nitrated, and this is also right. Benzene is also a stable material, but under harsh conditions is undergoes nitration, while graphene does not. This is not a critical point, though, I think that while I have doubts, I can still suggest to leave the question open and the paper being published.

Response: It’s our negligence for inaccurate writing, and we have deleted ‘Due to the stable structure of graphene sheet’ in Page 2 Line 6. The direct nitrification of graphene is hard to realize, but it may not ascribe to the stable structure of graphene sheet.

- another minor suggestion: "nitro connects" needs correction, because it is a "nitro group".

Response: It’s our negligence for inaccurate writing, and we have added ‘group’ in Page 2 Line 5.

This manuscript is a resubmission of an earlier submission. The following is a list of the peer review reports and author responses from that submission.

Round 1

Reviewer 1 Report

The authors used many different experimental techniques and computation analysis to describe the structure and properties of nitrated graphene (NG) as possible energetic material. I found the manuscript very difficult to read, not only for the poor English but also for the not clear exposition of the authors ideas and thoughts. Thus, the paper cannot be accepted in the present form for Nanomaterials. Below are few comments which might help to improve the quality of the manuscript for a resubmission to a different journal.

  1. The introduction is not clear, the authors discuss graphene oxide and then the use of nitrification for explosives but then they lack to provide a view on what can be the use for such different compounds and what will be gained for the use of such graphene derivatives. As it is now, it is difficult to see the novelty of the study, and this should be stressed more.
  2. The authors used a vast variety of techniques to characterize the investigated system, but the method section is somehow lacking of details. As example, in the computational part is not clear if the authors performed optimization or single point calculations, or why the distances were constrained (a rationale is missing). Or in the 'sample preparation' section, why the authors consider deionized water if then in the analyzed system they use a different solvent. An so on and so forth. 
  3. the 'Result and discussion' section is lacking of a real discussion and is difficult to read. Instead, the authors merely describe the obtained results without insights. Some examples are reported below.
  4. In section 3.1, it is not clear if the authors refer to a single layer NG or a few layers NG. How can be a single layer NG 1.1nm tick? It is either a multilayer or double layer NG. 
  5. In section 3.2 is reported that both nitro and nitrate groups are present, yet in the DFT section only the nitro are considered. Why?
  6. Section 3.3 is very hard to read. It is not clear, for example, how the different distances are treated. Are these lengths coming from a scan? Why the authors decided to use different lengths? Are these length coming from DFT or MD calculations? 
  7. As the charges are a key parameter in the analysis in section 3.4, more details should be reported. How the charges have been obtained? which scheme was used? To be sure the MD are significant, the density should also be considered. Once more, it is not clear why the authors consider water as a solvent, why they report that 'o-DCB and NMP can be regarded as excellent solvents'. MD simulations should have been performed with these two solvents. Also the authors should explain in more details the link between hydrophobicity and energy storage, as it is only mentioned in the submitted version.
  8. Section 3.5 reports the results of the thermal analysis, but it is not clear why NG is a better compound than GO. More details on the advantage of having NG should be provided. 
  9. Section 3.6 should be expanded. It is very interesting, but is a bit cryptic for a reader which is not in the field. The differences between GO and NG should be highlighted. Is GO used as retardant and NG to speed up the combustion? What are the possible application of NG?

Reviewer 2 Report

The submitted manuscript demonstrates the preparation of nitrated graphene by nitrification of graphene oxide. Various material characterization technique as well as theoretical studies were used to investigate the properties of the nitrated graphene. The manuscript is well written and characterization of the material is systematically performed. The work is interesting and can be accepted after the following comments are addressed:

  1. The term ‘’nitrated graphene’’ is inappropriate since the elemental analysis shows that after nitrification the graphene contains ~33.9% of oxygen which is similar to the oxygen content of GO before nitrification. Therefore, the authors should consider naming the material as ‘’nitrated graphene oxide’’ instead of ‘’nitrated graphene’’.
  2. Authors should provide the XPS analysis of the nitrated graphene in order to have a clear picture of bonding between C-N and types of nitrogen functional groups present in the graphene after nitrification.

Reviewer 3 Report

This is an interesting manuscript submitted by Guan et al. to be published in Nanomaterials. It reports on the synthesis and thermal/combustion behaviour of nitrated graphite oxide. Nitrated graphene is not a valid term, the carbon material still contains a lot of oxygen after the Nitrogen-modification. This should be reflected in the title. Also, in the title there is a mistake „two dimension” instead of „two-dimensional”. This fact and other shortcomings also shows that the English of the manuscript was not checked by the six co-authors with due care. This lowers the value of the submitted manuscript.

I think that the topic is important and interesting, nitrogen modification of graphene materials is an interesting aim, and it is well justified regarding the well-described motivation to formulate energetic materials. However, I can only suggest that the manuscript is further considered for publication if (1) the language is checked carefully by the authors and, more importantly, (2) they can give some convincing explanation why the nitration happens in the circumstances they use. I understand that the experimental results signify the presence of nitrogen in the samples, but do they really belong to the structure? If so, why does not the same reaction happen for pure graphite when the Hummers method is used, and there is no nitrogen incorporated into the GO sample already upon synthesis? The Hummers reaction mixture also contains the the same nitrate salt and cc. sulfuric acid! Since I guess the (NO2)+ cations should originate from the double protonation (and subsequent water elimination) of the nitrate nion by H2SO4 (which indeed happens in organic nitration reactions as the authors mention), why does it happen here and not during the GO synthesis?

For publication, this crucial issue must be clarified correctly.

Mistakes in „featured applications”:

  • „a novel two-dimension materials” is incorrect grammarly. „a” means singular case, „materials” is plural. The authors need to decide if they want to use a single or plural case.
  • This sentence makes no sense: „Combine the calculational optimization of the structure and the characterization of function groups to determine the nitrification progress that nitronium cation prefer to react with hydroxyl and epoxy groups in the edge”
  • „behaves the hydrophobicity” makes no sense

Science issues:

  • the „two-component structural model [9]” has a wrong reference associated. This is a paper by Dimiev et al. The authors should check the whole reference list for any mistaken assignment of references.
  • „amine nucleophiles undergo ring-opening reaction with epoxy groups [15]”. The cited reference is correct, it is specially focused on nucleophilic substitution on GO. However, the first related study on amine grafting by Bourlinos et al., Langmuir, 2003, could more justified to be cited (https://doi.org/10.1021/la026525h). The same is true if the cited papers are not the first reports on the specified organic linkers.
  • „2,4,6,8,10,12-hexanitro-2,4,6,8,10,12- hexaazatetracyclo (HNIW)”: this formulation is missing something important! There is not even „W” in the original name!
  • GO was prepared by the Hummers method. My question is, were there any modifications relative to the cited original paper? If yes, the authors should mention that, at which points were there changes (e.g. mass of reactants, temperature, heating rate, purification method, etc.) so that readers may be able to reproduce the results.
  • The calculated chemical formula looks completely wrong for GO. If the mass fraction of GO is 68.8/30.6/0.6 (C/O/H), then how can the chemical formula be C48H5O16? In the latter, there are 3* more Carbon than Oxygen, but in the mass ratio expression, C/O is only a bit larger than 2.
  • The value „2” in the upper index in the second line of 3.3 (C-NO2) and O-NO2 should be put into lower index, where the stoichiometry numbers are normally located.
  • What is the role of glacial acetic acid in the modification procedure? There should be mention about that in the manuscript.

Language issues:

  • „winkles” should be wrinkes
  • …was used to nitrite GO???
  • „and then tear off the filter membrane was teared off carefully” does not make sense